# Comparative Review of Cardioprotective Potential of Various Parts of *Sambucus nigra* L., *Sambucus williamsii* Hance, and Their Products

**DOI:** 10.3390/ijms27010460

**Published:** 2026-01-01

**Authors:** Beata Olas

**Affiliations:** Department of General Biochemistry, Faculty of Biology and Environmental Protection, University of Lodz, Pomorska 141/3, 90-236 Lodz, Poland; beata.olas@biol.uni.lodz.pl; Tel./Fax: +48-42-6354485

**Keywords:** cardiovascular disease, elderberry, phenolic compounds, *Sambucus nigra* L., *Sambucus williamsii* Hance

## Abstract

The genus *Sambucus* L. consists of about 29 recognized species (including 7 different genera that have berry fruit) distributed in all regions of the world. The most popular species are *Sambucus nigra* L., *Sambucus cerulean* Raf., and *Sambucus javanica* Blume, of which the European elderberry is widely used commercially. *S. williamsii* Hance (commonly known as *Jiegumu*) is endemic to China and is a valued variety of elderberry. *S. nigra* L. is one of the oldest medicinal plants. The herbal materials used in treatment and nutrition are its fruits, flowers, roots, leaves, and bark. Various parts of *S. williamsii*, including its fruit, flower, root, leaf, and stem, are also specifically used in Traditional Chinese Medicine. Additionally, berries and flowers of *S. nigra* L. have become a very popular inclusion in supplements, beverages, and foods in recent years. It is important that *Sambucus* plants are rich sources of various bioactive compounds, which determine their biological activities, such as antioxidant, antidiabetic, antimicrobial, and anti-inflammation. However, one of the most extensively studied species is *S. nigra* L. Among the different parts of this plant, the fruits and flowers are of particular interest due to their rich bioactive components. The aim of the present review is to provide and compare an overview of the cardioprotective potential of various parts not only of *S. nigra* L., but also of *S. williamsii* Hance, and their products in various models. Moreover, cardioprotective mechanisms of their main chemical constituents were demonstrated in this paper to provide a basis for further study and development.

## 1. Introduction

The genus *Sambucus* L. consists of about 29 recognized species (including 7 different genera that have berry fruit) distributed in all regions of the world and throughout most of the temperate and subtropical regions, for example, Europe, Asia, North America, North Africa, East and Southeast of Australia, and others. The most popular species are *Sambucus nigra* L. (its common names include black elder, elder, elderberry, European elderberry, European elder, and European black elderberry), *Sambucus cerulean* Raf. (blue elderberries), and *Sambucus javanica* Blume (Chinese elder), of which the European elderberry is widely used commercially. Moreover, *S. nigra* L. has three subspecies: *S. nigra* L. ssp. cerulea, *S. nigra* L. ssp. canadensis, and *S. nigra* L. nigra. It naturally occurs in most of Europe and in North Africa (Tunisia, Morocco, and Algeria) [1,2,3,4,5,6].

*S. williamsii* Hance (commonly known as *Jiegumu*) is endemic to China and is a valued variety of elderberry. It is also referred to as *Jiegudan*, *Qianqianhuo*, *Xugumu*, Tiesusan, and *Maniasosao*. In addition, *Sambucus ebulus* L. (dwarf elder) and *Sambucus sieboldiana* L. (Japanese red elder) are also often highly investigated [1,2,3,7,8,9,10].

Twelve *Sambucus* species have been used in ethnomedicine in various countries, including China, Iran, Turkey, Korea, and other countries. They have been applied in the treatment of herpes, chills, asthma, swelling, sinusitis, toothache, dropsy fever, headache, and others [4,11,12]. The traditional medicinal uses of *Sambucus* species involve various plant parts, including fruits, leaves, bark, flowers, and stems. For example, the berries and flowers have protective and therapeutic actions on diseases such as diabetes, constipation, cold, and flu. In addition, they have diuretic, catarrhal, and circulatory properties. The bark and leaves were found to have laxative, diuretic, and emetic actions. They are often applied in the treatment of diarrhea, rheumatism, cold, stomach aches, and constipation [7,13]. It is interesting that different parts of the *Sambucus* species have not only traditional medicinal applications, but also culinary applications. For example, their berry and flower extracts are an important component of various traditional drinks, including non-alcoholic and alcoholic beverages, tea, yoghourt, and ice cream. In addition, cakes and jams are also made from the berries of the *Sambucus* species. Their shoots and leaves may be used as vegetables when cooked [7,14,15,16].

*S. nigra* L. is one of the oldest medicinal plants. The herbal materials used in treatment and nutrition are its fruits, flowers, roots, leaves, and bark. Various parts of *S. williamsii*, including its fruit, flower, root, leaf, and stem, are also specifically used in Traditional Chinese Medicine. Additionally, berries and flowers of *S. nigra* L. have become a very popular inclusion in supplements, beverages, and foods in recent years [9,17,18]. Popular applications include syrups made from elderberry juice and tonics made by soaking the fruits in alcohol or water, freeze-dried materials, and the preparation of extracts for use in supplements. Elderberry products may also include tablets, lozenges, and gummies (as health supplements), and powdered fruits (as part of a drink mix). Moreover, they can be found as an ingredient in different food applications, including smoothies, energy drinks, kombuchas, juice, teas, and wine. Berries of *S. nigra* L. are also often used to make jellies and jams. Additionally, they can be used as a natural food dye in yogurt, kefir, meat products, and baked goods [17,19]. It was also shown that *S. nigra* L. flowers are natural flavouring components in non-alcoholic and alcoholic beverages, tea, and other food products, like ice cream and yoghurt [20]. More details about the application of *S. nigra* L. berries and flowers have been described in the review papers by Młynarczyk et al. [1] and Uhl and Mitechell [18]. It is interesting that *S. williamsii* has a taste profile characterized by a flat taste, sweetness, and bitterness [9].

*Sambucus* plants are rich sources of various bioactive compounds, which determine their biological activity, such as antioxidant, antidiabetic, antimicrobial, and anti-inflammation [7,21,22,23,24,25,26]. However, one of the most extensively studied species is *S. nigra* L. Among the different parts of this plant, the fruits and flowers are of particular interest due to their rich bioactive components [27,28].

It is known that *Sambucus* plants are commonly appreciated for various potential pharmacological properties, such as immunostimulant, anti-cold, anti-flu, and also antiviral, antibacterial, or anticancer. Moreover, the potential anti-cold, anti-flu, and immunostimulant actions result in the popularity of *Sambucus* in dietary supplements; however, strong scientific evidence of their efficacy is still lacking. Therefore, for the first time, the aim of the present review is to provide and compare an overview of the cardioprotective potential of various parts not only of *S. nigra* L., but also of *S. williamsii* Hance, and their products in various models. In addition, cardioprotective mechanisms of their main chemical constituents were demonstrated in this paper to provide a basis for further study and development. Potential molecular mechanisms of cardioprotection by chemical compounds from various parts of *S. nigra* L. may include its antioxidant activity (measured by different markers of oxidative stress), anti-inflammatory activity (by decreasing the level of nitric oxide and interelukin-2), lowering blood pressure (acting as a renin inhibitor), hypolipidemic activity, and anti-obesity potential (by increasing hepatic AMP-activated protein kinase).

## 2. Research Methods

The published literature about *S. nigra* L., *S. williamsii* Hance, their products, and their main chemical compounds with cardioprotective properties was collected from various scientific databases (for example, ScienceDirect, Web of Science, SCOPUS, Web of Knowledge, PubMed, Elsevier, Google Scholar, and Sci Finder). The search terms comprised the terms “*S. niger*”, “*S. williamsii*”, and “cardioprotection” and their combinations. No time criteria were applied to the search, but recent papers were evaluated first. Papers were first selected based on their relevance to the title of the present manuscript, and the identified articles were screened by reading the abstract. Any relevant identified articles were summarized. The last search was run on 6 December 2025. About 276 articles were obtained from the searches, and only 111 were included in this review. The data extracted from each article were the following: the type of studied material (species, cultivar, solvents used for extraction, or other relevant data), methods used in the study/study design, number of replicates, the cardioprotective activity, and statistical significance. If the P values were not provided for individual data points, it was assumed that the results were significant, unless it was stated otherwise in the text.

The cardioprotective potential of various parts of *S. nigra* L., *S. williamsii* Hance, and their main chemical compounds in various *in vitro* and *in vivo* models was summarized, and current studies are discussed. Moreover, the collected data also provided insights into the potential use of *Sambucus* preparations, especially *S. nigra* fruits and flowers, in functional foods and supplements with promising cardioprotective potential.

## 3. Phytochemical Characteristic of Various Parts of *S. nigra* and *S. williamsii*

### 3.1. S. nigra

Various parts of *S. nigra* are a source of different chemical compounds, but their chemical composition depends on many factors, including environmental conditions, variety, processing method, and storage conditions. For example, the flowering period is important for flowers, and the degree of ripeness is important for fruits. The diversity in the chemical composition of different parts of *S. nigra* applies in particular to phenolic compounds, including flavonols and anthocyanins, and vitamins, especially vitamin C [29]. For example, 1 cup (145 g) of *S. niger* fruits is considered an excellent source (containing > 20% US Daily Value) not only of vitamin C (40%), but also dietary fiber (25%) [18].

#### 3.1.1. Fruits

The fruits *of S. nigra* ripen at the turn of August and September, and their harvest is carried out in September and October. The colour of the fruit depends on the stage of its ripeness: at the beginning, the fruit is green, and when fully ripe, it is black and shiny [29].

The fruits *of S. nigra* contain about 80% water. The protein content of fresh fruit is approx. 2.8 g/100 g. This protein includes sixteen amino acids, nine of which are essential. Alanine, aspartic acid, and glutamic acid were identified as the dominant amino acids [1,2,3,29]. The fruit also contains about 18 g of sugars (including about 7.4 g of dietary fiber, especially pectin, protopectin, Ca-pectase, cellulose, pectin acid, and hemocellulose). The composition of sugars also includes simple sugars (especially glucose and fructose: 6.8–11.5 g of total sugars) [1,2,3,29]. Results of Veberic et al. [30] indicate that the content of total sugar ranged from 68.5 to 104.1 g/kg, depending on the selection and the cultivar. Authors also identified small amounts of sucrose (0.5–1.7 g/kg FW).

Fats are especially accumulated in the seeds of *S. niger* fruits (fat content—22.4%) and seed flour (fat content—16.0%). The major fatty acids are polyunsaturated fatty acids (75.1% of total fatty acids in seeds and 21.5% of total fatty acids in seed flour). For example, oleic acid (36.6 mg/kg fruits), linoleic acid (106 mg/kg fruits), and α-linolenic acid (61 mg/kg fruits) are present in the highest concentrations in seeds [1,2,3,20,29,31].

Organic acids constitute 1.0–1.3% of the fruit content, and citric acid was the most abundant. *S. nigra* fruits also include essential oils (about 0.01%), among about 53 chemical compounds, phenyl aldehydes (3.0–25.8% of the oil composition), and furfural (18%) predominate [1,2,3,29,30,31,32,33]. Moreover, lectins (about 0.1%) have been identified in *S. nigra* fruits. The mineral content (including Na, Ca, K, P, Fe, Mg, Mn, Zn, and Cu) represents 0.9–1.5% of the fruit mass. Moreover, *S. niger* fruits contain heavy metals such as cadmium and lead. Vitamin C content (6.0–132.1 mg/100 g) depends on the cultivar and the location. In addition, seed flour is a source of α-tocopherol. Other vitamins present in *S. nigra* fruits are vitamin B_2_, vitamin B_6_, folic acid, and biotin [1,2,3,20,29,34,35].

Interestingly, *S. nigra* fruits are an important source of phenolic compounds. In addition, their content in these fruits (364–1336 mg/100 g of fruit) is relatively high in comparison to other fruits. For example, the concentration of phenolic compounds in blackberries is about 248 mg/100 g of fruit, and in blueberries is about 525 mg/100 g of fruit [1,2,3,36,37,38,39]. The main group of phenolic compounds is anthocyanins (87.8–1816 mg/100 g of fruit), including cyanidin-3-glycoside, cyanidin-3-sambubioside, and cyanidin3-diglycoside [38]. The content of anthocyanins often increases as fruits ripen. Moreover, their content is dependent on the growing season [1,2,3,32,36]. Sources of anthocyanins are also various food products from *S. nigra* fruits, including juices, jams, and wines. For example, juices contain about 200 mg of anthocyanins/100 mL, while juice concentrate, about 411 mg/100 mL, and the predominant anthocyanin is cyanidin-3-sambubioside. Cyanidin-3-glucoside was identified as the most abundant anthocyanin in *S. nigra* fruit wine [1,2,3,40,41]. However, of the literature published on the analysis of anthocyanins using conventional methods, including HPLC, only 2% was published on *S. nigra* fruits [42].

Recently, results of Terzic et al. [42] have demonstrated that the contents of other phenolic compounds in *S. nigra* fruit wine (per mL) are also high (for example, 43.3 µg of quercetin, 17.7 µg of chlorogenic acid, and 52.5 µg of protocatechuic acid). Moreover, *S. nigra* fruit pomace is also very rich in anthocyanins, and it contains 75–98% of total anthocyanidins found in fresh fruits [1,2,3]. On the other hand, anthocyanidins are sensitive to different environmental factors, including light, temperature, pH, and others [43,44].

Various other phenolic compounds, including phenolic acids and flavonoids, were also identified in *S. nigra* fruits. Among phenolic acids, chlorogenic acid (10–32 mg/100 g of fruit) is an important acid. Other important phenolic acids are gentisic acid (2.2 µg/100 g of dry weight) and gallic acid (3.4 µg/100 g of dry weight). However, among flavonoids (approx. 186 mg/100 g of fruit), particular attention should be paid to quercetin (228.8 µg/100 g of dry weight) and rutin (813.1 µg/100 g of dry weight) [1,2,3,29,34]. It is worth noting that the content of anthocyanins and other phenolic compounds in various food products from *S. nigra* fruits depends on the technological conditions used or the degree of processing [1,2,3,29]. For example, various cooking methods, including baking, boiling, and steaming, could lead to a reduction in their content, and the recipe without thermal treatment could maximally preserve anthocyanin content [34,45]. Moreover, different processing techniques can also affect the stability of these bioactive compounds in *S. nigra* fruits [46,47].

In addition, Kaack et al. [32] observed that *S. nigra* fruit juices (processed using enzymatic treatment) have a lower average content of phenolic compounds compared to the juices produced without enzymatic treatment. Alcoholic fermentation of *S. nigra* fruits also causes changes in the content of various phenolic compounds. Results of Młynarczyk and Walkowiak-Tomczak [2] indicate that the hot-pressed *S. nigra* fruit juices have higher total content of bioactive compounds, including phenolic compounds (for example, anthocyanins), than the cold-pressed juices. In another paper, Młynarczyk and Walkowiak-Tomczak [3] evaluated the bioactive properties of selected commercial *S. nigra* products made from fruits and flowers (syrups, juices, jam, confiture, and mousse). The evaluation was based on measurements of antioxidant properties, total phenolic compounds, as well as anthocyanin content and profile. For example, the highest values of these parameters were shown by one of five juices, which was pressed from organic fruits.

More details about the chemical composition of the three subspecies of *S. nigra* fruits (*S. nigra* ssp. *nigra*, *S. nigra* ssp. *canadensis*, and *S. nigra* ssp. *cerulea*) are described in the review paper of Uhl and Mitchell [18]. For example, *S. nigra* ssp. *canadensis* fruits have a similar composition of phenolic compounds, but a lower concentration of anthocyanidins as compared with *S. nigra* ssp. *nigra*. In addition, a few unique phenolic compounds, including protocatechuic acid dihexoside (14.4–57.0 g/100 g FW) and 5-hydroxypyrogalloll (24.5 ± 22.4 g/100 g FW), have been identified only in *S. nigra* ssp. *cerulea* [18]. However, there are currently only a few or no available data on the macronutrient and micronutrient content of *S. nigra* ssp. *canadensis* and *S. nigra* ssp. *cerulea* [18].

Recently, Onolbaatar et al. [48] have compared the phytosterol and triterpenoid profiles of fruits and fruit-derived products of selected *Sambucus* plants, including *S. nigra* and *S. racemose*. They observed that elderberry fruits are rich sources of phytosterols, especially sitosterol (which represented up to 88% in red elder *S. racemose*), and triterpenoids (13.81 mg/g in *S. nigra* f. *porphyrophylla* cultivar Black Lace “Eva”). The phytochemical analysis showed that ursolic acid and oleanolic acids are the main triterpenoids in the extract from *S. nigra* fruits [48,49]. Among various fruit-derived products (jam, juice, syrup, and wine), jams had the highest concentration of phytosterols and triterpenoids (a total of 340 µg/g) [48].

#### 3.1.2. Flowers

The literature data indicate that 100 g of *S. niger* flowers contains about 2–2.5% protein, various vitamins (vitamins B, vitamin A, and vitamin C (80–90 mg/100 g)), or minerals. They are also a source of various phenolic compounds, including phenolic acids (e.g., chlorogenic acid accounts for about 35% of all phenolic compounds), and flavonoids, such as quercetin (10% of all phenolic compounds), kaempferol, rutin, isoquercitrin, or astragaline. Other groups of phenolic compounds are flavanones, including naringenin, and flavanols, such as epicatechin, catechin, and procyanidin trimer [50,51]. In addition, fresh flowers are a source of triterpenes (oleanoic acid, ursolic acid, and α- and β-amarin) and phytosterols (β-sitosterol, stigmasterol, and campesterol). *S. nigra* flowers also include essential oils (about 58 compounds) [1,2,3,29]. However, the content of compounds in *S. nigra* flowers depends on storage conditions, soil conditions, climate, or the method of transporting the raw material for processing [1,2,3,8,29].

#### 3.1.3. Leaves

Chemical analysis has found 100 g *S. nigra* leaves contain 3 g proteins, 200–3000 mg vitamin C, tannins, alkaloids, organic acids, aldehydes, or sluices [29]. The protein concentration is higher in the leaves than in the flowers [8].

#### 3.1.4. Bark

It is important to note that the inedible part of *S. nigra*, its bark, is also abundant in biologically active compounds. However, the chemical composition of the bark is not fully understood, but it contains a certain amount of tannins. In addition, it contains glycosides, alkaloids, choline, saponins, essential oils, and sitosterols, including β-sitosterol [8,29].

### 3.2. S. williamsii

About 238 chemical compounds, including sugars, phenolic compounds, alkaloids, terpenoids, and other components, have been identified in various parts of *S. williamsii*, such as its fruits, leaves, stems, bark, and root. For example, aldehydes are the predominant constituents of *S. williamsii* oil. Moreover, 14 trace elements (Ca, K, Zn, Ba, Fe, Al, V, Cr, Ti, Cu, Sr, Mn, P, and Ni) and 17 amino acids (for example, serine, threonine, glycine, cysteine, alanine, valine, leucine, and others) also have been isolated from *S. williamsii* fruits, which contribute to the nutritional characteristics of its fruits [52,53]. In addition, various phenolic compounds have been identified in *S. williamsii* fruits. Stems and branches also are a source of phenolic compounds, for example, vanillin, acetovanillone, 4-hydroxy-benzoic acid, protocatechuic acid, and others [54]. Moreover, 34 terpenoids (sequiterpenoids, iridoid glycosides, and triterpenoids) have been identified in these parts of *S. williamsii* [55]. More details about the phytochemistry of *S. williamsii* are described by Lei et al. [9].

Main chemical compounds with cardioprotective potential of various parts of *S. nigra* and *S. williamsii* are demonstrated in Table 1.

## 4. Cardioprotective Potential of Various Parts of *S. nigra*, *S. williamsii*, and Their Products (*In Vitro* and *In Vivo* Models)

Reviewing the literature indicates that various plant food products, like tea, cocoa, coffee, and others, exhibit cardiovascular protective action using many mechanisms of action [56,57,58,59]. Different health benefits associated with the consumption of berries are also demonstrated in epidemiological effects on CVDs. For example, various berries (such as sea buckthorn berries (*Hippophae rhamnoides*), aronia berries (*Aronia melanocarpa*), goji berry (*Lycium barbarum*), and others in forms of fresh berries, wine, juice, teas, and supplements) may play an important role in modulating hemostasis, including blood platelet functions [60,61]. However, the cardioprotective potential of berries is dependent on a range of factors, such as the type of berries, the form of consumption, and their bioactive composition, especially phenolic compounds: anthocyanidins, procyanidins, flavonols, and phenolic acids [62]. For example, these phenolic compounds may modulate blood platelet function by modifying the expression of blood platelet receptors and changing the activity of signaling enzymes. They also change the level of ROS [38]. More details about the implications of berries for human health are described in a few review papers [28,38,60,61].

Studies also indicate that *S. nigra* and *S. williamsii* have a protective action on the cardiovascular system in *in vitro* and *in vivo* models. For example, all parts of *S. nigra,* flowers, fruits, leaves, bark, and its food products, contain numerous bioactive compounds, mainly with antioxidant activities, which may determine its health-promoting properties, including cardioprotection action. For example, anthocyanins reduced risk factors for cardiovascular diseases (CVDs), including obesity associated with CVDs [38,63,64]. In addition, anthocyanin mixtures found in food, such as *S. nigra* fruits, had higher clinical efficacy than single anthocyanins [38].

### 4.1. S. nigra

Oxidative stress has been implicated in the pathogenesis of various cardiovascular diseases, including ischemic heart disease, hypertension, and others [22,65,66,67,68]. On the other hand, different plant preparations, including food products and supplements, have antioxidant potential. Studies based on various techniques, such as the 2,2-diphenyl-1-picrylhydrazyl (DPPH) radical scavenging method, the oxygen radical absorbance capacity (ORAC), and others, found that various parts of *S. nigra* and *S. williamsii* have antioxidant properties and that the key antioxidants are phenolic compounds that can help to reduce oxidative stress. For example, several *in vitro* studies indicate the antioxidant capacity of *S. nigra* extracts, especially extracts from fruits, based on DPPH and ORAC [69,70]. Młynarczyk et al. [1] also observed that *S. nigra* fruits are characterized by high antioxidant properties, which range from 82.1 to 89.2% of inhibition in relation to the DPPH radical. On the other hand, Duymus et al. [70] noted that the extract from *S. nigra* fruits has lower antiradical activity (towards DPPH) than ascorbic acid used as a standard. The IC_50_ value was 123 µg/mL for water extract, while it was 8 µg/mL for ascorbic acid. The results obtained by Espin et al. [71] demonstrated that commercial concentrates of *S. nigra* fruits have lower radical scavenger capacity towards DPPH compared to other tested sources of anthocyanins (for example, extracts from strawberries or blackthorn fruits).

Młynarczyk and Walkowiak-Tomczak [2] studied the antioxidant properties (using the ABTS (2,2′-azino-bis(3-ethylbenzothiazoline-6-sulfonic acid) radical) of *S. nigra* juices according to the fruit origin and cultivar, as well as various conditions of juices processing and preservation (with or without pasteurization). Juices were made from wild *S. nigra* fruits and from “Samyl”, “Haschberg”, and “Sampo” cultivars grown at plantations. Four variants of juice were prepared: cold-pressed, hot-pressed, pasteurized, and unpasteurized. The authors observed that the hot-pressed juices have higher antioxidant properties than cold-pressed juices.

It is interesting that *S. nigra* flowers often have higher antioxidant properties than other parts of this plant, such as leaves and fruits. For example, Kołodziej and Drożdżal [72] compared antioxidant properties of aqueous extracts of *S. niger* flowers and fruits harvested in 17 wild sites in Poland, and the effect of total phenolic compound content on antioxidant properties of the raw material was studied. They observed that *S. nigra* flowers contained a higher number of phenolic compounds than the fruits harvested in the same wild sites of their occurrence, and their content significantly depended on the wild site where they were harvested. In addition, the results proved a positive correlation between the content of phenolic compounds and the antioxidant properties of *S. nigra* raw material. Moreover, authors observed that *S. nigra* flowers have stronger DPPH radical inhibition activity (91.9–94.1%) in comparison to leaves (16.8–48.5%) and fruits (50.2–67.7%). In addition, according to Stoilova et al. [73], *S. nigra* flowers extract possesses greater radical scavenging properties in comparison to rutin.

The literature data indicate that *S. nigra* extract (15 mg/kg of bw) attenuates oxidative stress and inflammation in femoral ischemia [74], which is a major cause of cardiovascular diseases, with an increased rate of morbidity and mortality, pathogenetically characterized by redox imbalance, inflammation, and tissue damage. The authors investigated the effects of *S. nigra* fruit extract on the gastrocnemius muscle lesions induced by the experimental femoral ischemia. For example, the tested extract demonstrated beneficial antioxidant effects, decreasing the lipid peroxidation (measured by the level of malondialdehyde (MDA)) in muscle homogenates. This extract also showed anti-inflammatory effects through reducing interleukin-6 (IL-6).

Waldebauer et al. [75] observed that the extract of the lyophilized pomace of *S. nigra* fruits (50 µg/mL) has a cardiovascular action through increasing A23187-stimulated endothelial nitric oxide synthase (eNOS) activity in human endothelium-derived cell line EA.hy926. Moreover, authors noted that the major effective chemical compounds of the tested extract were di- and trihydroxylated triterpenic acids.

It was also found that salicylic acid (aspirin) is present in *S. nigra* fruits. This compound has long been used as an anti-inflammatory and antiplatelet agent. However, the use of salicylic acid for the prevention of CVDs is limited because its use increases the bleeding risk [10].

In addition, various preparations from *S. nigra* fruits also exhibited cardioprotective potential in *in vivo* animal models. For example, Dubey et al. [76] found that supplementation with anthocyanin-rich *S. nigra* extract reverses lipid peroxidation observed with dietary fish oil alone in BioF1B hamsters. Both liver and plasma thiobarbituric acid reactive substances (TBARSs) showed significant reductions upon supplementation with *S. nigra* extract in fish oil-fed BioF1B hamsters. Their findings also demonstrated that supplementation with *S. nigra* extract reverses hyperlipidemia.

The study of Ciocoiu et al. [77] investigated the effects of the association between the renin inhibitor and the polyphenolic extract from *S. nigra* fruits (a dosage of 0.046 g/kg body weight, every 2 days, for 8 weeks) on biochemical parameters and systolic and diastolic blood pressure within a Nω-nitro-L-arginine methyl ester (L-NAME)-induced experimental model of arterial hypertension in the Wistar white rats. They found that the total antioxidant capacity levels were significantly decreased in the group with arterial hypertension as compared to the control rats. A combination of a renin inhibitor (Aliskiren) and a tested fruit extract generated a superior antioxidant effect compared to administering the two separately. In addition, both systolic and diastolic pressure in rats with drug-induced hypertension were reduced by *S. nigra* extract.

The objective of the study of Farrell et al. [78] was to determine whether an anthocyanin-rich *S. nigra* fruit extract (13% anthocyanins) would protect against inflammation-related impairments in high-density lipoprotein (HDL) function and atherosclerosis in apoE(−/−) mice (a mouse model of hyperlipidemia and HDL dysfunction). The 10-week-old male apoE(−/−) mice were supplemented with 1.25% (*w*/*w*) *S. nigra* extract or a control diet for 6 weeks. After 6 weeks, serum lipids did not differ significantly between groups, while aspartate transaminase and fasting glucose were reduced in mice treated with plant extract. Moreover, hepatic and intestinal mRNA changes with tested extract-feeding were consistent with an improvement in HDL function (Apoa1, Pon1, Saa1, Lcat, Clu) and a reduction in hepatic cholesterol levels. In tested extract-fed mice, serum paraoxonase-1 (PON1) activity was significantly higher. Authors also observed significant reductions in total cholesterol content of the aorta of BEE-fed mice, indicating less atherosclerosis progression.

In another *in vivo* model, Millar et al. [79] studied the long-term consumption of *S. nigra* fruit extract on HDL function and atherosclerosis in apolipoprotein (apo) E(−/−) mice. ApoE(−/−) mice (n = 12) were fed a low-fat diet and supplemented with 0, 0.25%, or 1% (by weight) *S. nigra* fruit extract (about 37.5–150 mg anthocyanins per kg body weight) for 24 weeks. Authors noted that chronic supplementation with the tested fruit extract in apoE(−/−) mice dose-dependently improved HDL function.

Mauray et al. [80] investigated the impact of *S. nigra* fruit anthocyanin-rich extract (0.02%, for 2 weeks) supplementation on gene expression in the liver of apo E(−/−) mice, the widely used model of atherosclerosis. Their results indicate that a 2-week supplementation significantly reduced plasma total cholesterol and hepatic triglyceride levels, whereas the plasma antioxidant status remained unchanged. Moreover, transcriptional analysis, using microarrays, revealed that the expression of 2289 genes was significantly altered. The tested plant extract over-expressed genes involved in bile acid synthesis and cholesterol uptake into the liver and down-regulated the expression of pro-inflammatory genes.

Only a few clinical trials have been conducted on anthocyanins extracted from *S. nigra* fruits and have demonstrated positive action on CVDs. For example, Curtis et al. [81] determined the effect of chronic consumption of *S. nigra* fruit anthocyanins on the biomarkers of CVD risk. In this experiment, 26 healthy postmenopausal women were administered 500 g/day of anthocyanins for 12 weeks. Their results showed that the chronic intake of anthocyanins from *S. nigra* fruits is safe, but does not affect the biomarkers of CVD risk.

Another randomized study indicates that the administration of low doses of lyophilized *S. nigra* fruit juice (50 mL/day, for one week) has a limited effect on lowering serum lipids, cholesterol, and triglyceride levels in 34 healthy volunteers [82].

Nillson et al. [83] evaluated effects on cardiometabolic risk markers (including, blood pressure, blood concentration of lipids, inflammatory markers, and markers of oxidative stress) of 5 weeks intervention with a mixture of berries (150 g blueberries, 50 g blackcurrant, 50 g elderberry, 50 g lingonberries, 50 g strawberry, and 100 g tomatoes, daily), in healthy humans (n = 40). The daily amounts of total phenolic compounds and fiber from the berry beverage were 795 mg and 11 g, respectively. Authors observed that the berry supplementation reduces total- and low-density lipoprotein (LDL) cholesterol compared to the control beverage (water-based).

Recently, results of Alqudah et al. [84] found that dietary *S. nigra* fruits extract abrogates the effects of an obesogenic diet in a gut microbiota-dependent manner by preventing insulin resistance and reducing hepatic steatosis in mice. In addition, authors suggest that hydrocinnamic acid (as a key microbial metabolite, enriched in the portal vein plasma of *S. nigra* fruit-supplemented animals) potently activates hepatic AMP-activated protein kinase α, explaining its role in improved liver lipid homeostasis.

Figure 1 demonstrates that *S. nigra* exhibits protection on the cardiovascular system and CVDs through various mechanisms, including suppression of oxidative stress, regulation of lipid metabolism and inflammation, action as renin inhibitors, and others.

### 4.2. S. williamsii

Only a few papers indicate that preparations from various parts of *S. williamsii* have a positive action on risk factors of CVDs. For example, Su et al. [85] analyzed the antioxidant properties of flavonoids in four polar extracts of *S. williamsii* leaves (water, ethyl acetate, n-butanol, and chloroform), but the ethyl acetate extract has the highest scavenging activity. Fang et al. [86] also reported that the 70% ethanol extract of *S. williamsii* fruits has free radical scavenging capacity *in vitro*.

In addition, Liu [34] extracted and purified bioactive components from various parts of *S. williamsii*. The antioxidant and anti-inflammatory properties were also studied *in vitro*. *S. williamsii* lignans exhibited superior anti-inflammatory activity, and *S. williamsii* anthocyanins had the highest antioxidant potential.

Xiao et al. [52] suggested that various phenolic acids, including vanillic acid and ferulic acid isolated from the stem and root bark of *S. williamsii,* possess therapeutic effects against CVDs. According to Xiao et al. [52], terpenoids isolated from *S. williamsii* have also cardioprotective potential.

Recently, Sun et al. [87] observed that saponins presented in *S. williamsii* leaves reduce oxidative damage in mice.

Results of Lv et al. [88] indicate that linoleic acid from *S. williamsii* seed oil (1–4 g/kg bw) has hypolipidemic properties. This activity of linoleic acid was investigated *in vivo* using hyperlipidemia mice models fed with the linoleic acid at doses of 1, 2, and 4 g/kg bw. The authors observed that serum lipid levels were highly significantly improved. In addition, the DPPH free radical scavenging assay was used to determine its antioxidant activity of linoleic acid. IC_50_ of DPPH radical scavenging activity of linoleic acid was 61.92 mg/mL. In this experiment, linoleic acid was extracted by a high-pressure fluid.

Xiao et al. [51,52] investigated how the lignan-rich fraction from *S. williamsii* modulates lipid metabolism in menopausal women. Their results demonstrated that oral administration of this fraction (140 mg/kg and 280 mg/kg) for 10 weeks alleviates dyslipidemia and improves liver functions.

There is also one article about the cardioprotective potential of *S. ebulus* ripe fruits [89]. In this paper, the authors established the effect of *S. ebulus* L. fruit consumption on body weight, blood pressure, lipid profile, and antioxidant markers in healthy volunteers (n = 21). Participants consumed 200 mL of *S. ebulus* fruit infusion/day for a period of 30 days. The authors observed a significant decrease in triglycerides, total cholesterol, and LDL. Moreover, the HDL/LDL ratio increased by about 43%. Improved serum antioxidant capacity and total thiol levels were also noted.

## 5. Toxic Action and the Bioavailability of Bioactive Compounds

It is important that all parts of *S. nigra*, especially its older parts, can be toxic, for example, when ingested as fruits in high amounts due to the accumulation of cyanogenic glycosides. For example, sumbunigrin, holocalin, prusasin, and zierin metabolites have been isolated from *S. nigra* fruits [90] and flowers [91]. The highest amounts of sambunigrin (27.7–209.6 µg/g FW) are present in elder leaves compared to other parts of this plant [92]. Cyanogenic glycosides are toxic and life-threatening because they can be hydrolyzed, resulting in the release of cyanide. On the other hand, they are degraded during heat treatment. For example, the concentration of sambunigrin was reduced from 18.8 mg/kg in unprocessed fruits to 10.6 mg/kg in juice, 2.8 mg/kg in tea, and 0.8 mg/kg in liqueur [93].

Other toxic compounds present in various parts of *S. nigra* are lectins (in fruit—nigrin f; in bark—nigrin b-SNA V, SNA I; and seeds (nigrin s)) [58]. In addition, *S. nigra* contains the allergen Sam n1 [94].

However, based on various data, it is believed that the fruits of *S. nigra* can be safely consumed when ripe, dried, or cooked [95]. Moreover, the results of the safety of *S. nigra* fruits and flowers demonstrated no toxic action or side effects on the human body. *S. nigra* fruits and flowers are recognized by the US Food and Drug Administration (FDA) as a safe food additive, which is indicated by the GRAS (Generally Recognized As Safe) status. On the other hand, the European Medicines Agency (EMA) points to the safety of their use, but recommends limiting their use by children under 12 years of age and women during pregnancy and lactation [95,96]. It is also important to separate the seeds from the pulp when preparing *S. nigra* fruit products. This is because the ingested seeds contain ingredients that react with enzymes contained in the stomach, which secretes hydrocyanic acid [95,96].

Moreover, flowers of *S. nigra* L. have been approved by the Commission E of the General Federal Institute for Drugs and Medical Devices to treat flu and cold. For example, various products (especially as supplements, including OptiBerry IH141, Sinupret^®^, Sambucol^®^, Tretussin^®^, Sambuca-Well^®^, and others) containing not only S. nigra flowers, but also fruits have gained considerable popularity in Europe and North America. Recently, Festa et al. [97] have noted that elderberries may be a potential supplement to improve vascular function in a SARS-CoV-2 environment.

In the food industry, flowers are mainly used in dried form, mainly for the production of herbal teas. They are also often used to make alcoholic liquors [1,2,3,18,29,34,98,99].

Recently, Ferreira-Santos et al. [99] studied the effect of gastrointestinal digestion on the toxicity of extracts from *S. nigra* fruits and flowers. The digested and non-digested extracts had different effects on various cell lines. For example, the IC_50_ values were highest for normal cell lines (L929), indicating low toxicity, while lower values were noted for cancer cell lines (HT29 and HeLa).

Metabolism of phenolic compounds, especially anthocyanins, after oral administration of various *S. nigra* berry products, including juice, is often investigated. However, anthocyanins demonstrate low bioavailability compared with other phenolic compounds [100,101,102]: the peak plasma concentration is reached 6 h after consumption, and these concentrations remain in the micromolar range. For example, Netzel et al. [102] observed that *S. niger* fruit juice (400 mL, containing 722 mg of total anthocyanins) causes the urinary excretion of about 113 µg/h of these compounds in healthy people. In this study, in eight healthy subjects receiving a single oral dose of this juice, the pharmacokinetic parameters were obtained from 7 h urine excretion.

Anthocyanins are metabolized by conjugation in the intestinal and hepatic [102,103,104,105,106]. Most significantly, toxicological studies support the view that anthocyanins pose no threat to human health [107], and they are safe for consumption, even at higher doses [108]. Moreover, anthocyanins are able to cross the blood-brain barrier [108].

On the other hand, no research is available comparing the bioavailability of bioactive compounds in various *S. niger* food products, especially tablets, liquid preparations, and gummy, or products based on elderflowers. In addition, the bioavailability of bioactive compounds from *S. nigra* differs based on the delivery mechanism. Moreover, *S. nigra* preparations may be more inherently stable than others. For example, ascorbic acid is not stable in gummy preparations [18].

Only a few studies have assessed the toxicity of different parts of *S. williamsii*. For example, Zhang et al. [109,110] demonstrated that the toxicity rates of *S. williamsii* leaves and stems are 10%, 30%, and 89%, when *S. williamsii* content in the diet reaches 10%, 15%, and 20%, respectively. In this experiment, the toxicity was studied in mice. Recently, results of Zhang et al. [110] have indicated that S. williamsii seed oil is cytotoxic when the mass concentration of its volatile components exceeds 0.1 g/L. Unfortunately, as there is not yet any clinical evidence for the safety of *S. williamsii*, further research is needed to confirm this. In addition, toxicology studies on various parts of *S. williamsii* can offer valuable insights for its edible and medicinal uses, especially in humans.

There are also only a few papers about the pharmacokinetics of *S. williamsii*. For example, the result of Weng [111] demonstrated that morroniside in *S. williamsii* is rapidly absorbed and eliminated in rats, resulting in low bioavailability.

## 6. Concussions

Even though various studies on *S. nigra* and *S. williamsii* have long concentrated on their bioactive compounds and biological properties, there are still numerous gaps in understanding their cardioprotective action, and pharmacological trials for the discovery of new supplements and medicines as a promising strategy for preventing cardiovascular diseases and promoting their optimal cardioprotective function. For example, especially flowers and fruits of *S. nigra* or its food products, including fruit juices, contain numerous bioactive compounds, mainly with antioxidant activities, which may determine its health-promoting properties, including cardioprotection action, making them a very valuable pharmaceutical raw material, especially for the production of supplements and functional foods. In addition, the available literature also indicates that these compounds have anti-inflammatory and anti-obesity properties. However, it is not correct to prove the beneficial effects of various parts of *S. nigra* on the prevention and treatment of CVDs in humans, especially in studies with a large sample size. Therefore, there is a need for more comprehensive evidence-based clinical studies and data.

In particular, *S. nigra* phenolic compounds play an important role in cardioprotective mechanisms. However, other bioactive compounds, such as fiber, unsaturated fatty acids, phytosterols, and others, may also play a positive role in the prophylaxis and treatment of CVDs. While it is significant that these bioactive components exist, their mutual interaction is largely unknown. Again, this matter needs further experiments.

In addition, not only *S. nigra*, but also *S. williamsii* food products with cardioprotective potential should be studied further. Such studies should also examine this potential for the long term. The reports on the underlying cardioprotective mechanisms of action and the bio-constituents of *S. nigra* and *S. williamsii* associated with this activity are also limited. Moreover, there are no clinical experiments for the interactions of *S. nigra* and *S. williamsii* preparations with different drugs and supplements used in the prophylaxis and treatment of CVDs. Despite this, for the first time, this review paper indicates that especially *S. nigra* fruits, flowers, and its fruit juice may be new good candidates as functional foods with cardioprotective properties. *S. nigra* fruits, flowers, and fruit juice may be good candidates for functional foods with cardioprotective properties. However, well-established long-term clinical trials in this context should be a hot topic in the future. In addition, clinical validation and addressing toxicity concerns would provide a more comprehensive perspective on their potential use.

## Figures and Tables

**Figure 1 ijms-27-00460-f001:**
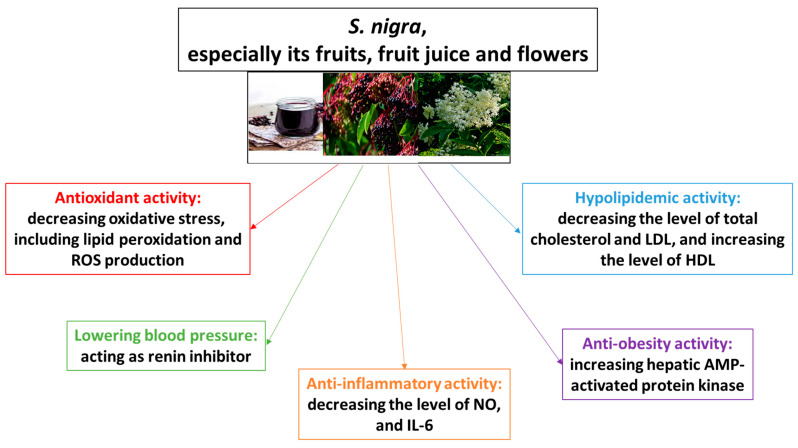
Potential molecular mechanisms of cardioprotection by various parts of *S. nigra* and its products.

**Table 1 ijms-27-00460-t001:** Main chemical compounds with cardioprotecive potential of various parts of *S. nigra* and *S. williamsii*.

Part of Plant	Chemical Compound				
	Phenolic Compounds	Dietary Fiber	Unsaturated Fatty Acids	Phytosterols	Terpenoid Compounds
** *S. nigra* **					
**Flowers**	+	−	−	+	+
**Fruits**	+	+	+	+	+
**Leaves**	−	−	−	−	−
**Bark**	−	−	−	+	+
** *S. williamsii* **					
**Flowers**	+	−	−	−	−
**Fruits**	+	−	−	−	−
**Leaves**	−	−	−	−	−
**Bark**	−	−	−	−	+

## Data Availability

No new data were created or analyzed in this study. Data sharing is not applicable to this article.

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
