# Peer review of "Int. J. Mol. Sci.2026, 27(1), 460;https://doi.org/10.3390/ijms27010460"

_ijms, 2026, doi:10.3390/ijms27010460_

Round 1

Reviewer 1 Report

Comments and Suggestions for Authors

The manuscript concerns cardioprotective potential of Sambucus nigra and Sambucus williamsi plant parts and products. It is a review based on data from scientific literature.

  1. The manuscript is clearly written and interesting, however, it is not obvious if it fits to IJMS. The mechanism of cardioprotective activity is described in a general way, more suitable for other MDPI journals, like for example Plants – particularly because the cited studies have been done on extracts, not on isolated compounds. However, such opinion is rather given by the Editor, not the Reviewer.
  2. The title, Comparative of cardioprotective potential.., seems to be not correct grammatically. Comparative is adjective, it requires a noun. So it should rather be: Comparative review of… or just Comparison of …
  3. The choice of the activity – i.e. cardioprotective – should be more clearly justified; it is not enough to write that some foods, like tea, cocoa or coffee exert such properties, and that some published papers mentioned this activity in case of Sambucus. This justification of choice is particularly of interest since Sambucus is commonly appreciated for other potential pharmacological properties, such as immunostimulant, anti-cold, anti-flu, and also antiviral, antibacterial or anticancer. (The potential anti-cold, anti-flu and immunostimulant action results in the popularity of Sambucus in dietary supplements, however, the strong scientific evidence of their efficacy is still lacking.)
  4. The Author focused mainly on phenolics, and it is understandable, since these compounds are usually regarded as responsible for the majority of health benefits of plant foods and products. However, in the part concerning phytochemical characteristics of Sambucus spp., it would be of interest to add more data on another group of valuable compounds, i.e. triterpenoids and phytosterols. These compounds are mentioned in the manuscript, but only as the ingredients of the flowers, whereas their presence in fruits is well known and of great interest according to their anticancer and anti-inflammatory activities (Gleńsk et al., E. Triterpenoid acids as important antiproliferative constituents of European elderberry fruits. Cancer 2017, 69, 643–651; and the recent study on the occurrence of triterpenoids and sterols in elderberry-derived foods: Onolbaatar et al. Fruit and fruit-derived products of selected Sambucus plants as a source of phytosterols and triterpenoids. Plants 2025, 14, 1490 - these references could be cited in the manuscript). The presence of triterpenoids, including triterpenoid acids, requires attention because these compounds can participate in cardioprotective activity by reducing oxidative stress, inhibiting apoptosis, and improving heart function. The Author mentioned that polyhydroxylated triterpenoid acids were regarded as probably the major effective compounds of extracts tested in one of the cited studies.

Author Response

The manuscript concerns cardioprotective potential of Sambucus nigra and Sambucus williamsi plant parts and products. It is a review based on data from scientific literature.

Response: Thank you for reviewing the manuscript and providing such helpful comments. All of them have been taken into consideration when revising the manuscript.

  1. The manuscript is clearly written and interesting, however, it is not obvious if it fits to IJMS. The mechanism of cardioprotective activity is described in a general way, more suitable for other MDPI journals, like for example Plants – particularly because the cited studies have been done on extracts, not on isolated compounds. However, such opinion is rather given by the Editor, not the Reviewer.

Response: I have decided to send my manuscript to IJMS, because Plants described very often only physiological processes in plants. Moreover, plant extracts have often bigger biological activity than isolated pure compounds.

  1. The title, Comparative of cardioprotective potential.., seems to be not correct grammatically. Comparative is adjective, it requires a noun. So it should rather be: Comparative review of… or just Comparison of …

Response: I have corrected the title of my manuscript. Now, it is: “Comparative review of cardioprotective potential of various parts of Sambucus nigra L., Sambucus williamsii Hance and their products”.

  1. The choice of the activity – i.e. cardioprotective – should be more clearly justified; it is not enough to write that some foods, like tea, cocoa or coffee exert such properties, and that some published papers mentioned this activity in case of Sambucus. This justification of choice is particularly of interest since Sambucus is commonly appreciated for other potential pharmacological properties, such as immunostimulant, anti-cold, anti-flu, and also antiviral, antibacterial or anticancer. (The potential anti-cold, anti-flu and immunostimulant action results in the popularity of Sambucus in dietary supplements, however, the strong scientific evidence of their efficacy is still lacking.)

Response: I have added more information about it (the chapter of Introduction): “It has known that Sambucus plants are commonly appreciated for various potential pharmacological properties, such as immunostimulant, anti-cold, anti-flu, and also antiviral, antibacterial or anticancer. Moreover, the potential anti-cold, anti-flu and immunostimulant action results in the popularity of Sambucus in dietary supplements, however, the strong scientific evidence of their efficacy is still lacking. Therefore, the first time, the aim of the present review is to provide and compare an overview of the cardioprotective potential of various parts not only of S. nigra L., but also of S. williamsii Hance, and their products in various models. Moreover, cardioprotective mechanisms of their main chemical constituents were demonstrated in this paper to provide a basis for further study and development.”

  1. The Author focused mainly on phenolics, and it is understandable, since these compounds are usually regarded as responsible for the majority of health benefits of plant foods and products. However, in the part concerning phytochemical characteristics of Sambucus spp., it would be of interest to add more data on another group of valuable compounds, i.e. triterpenoids and phytosterols. These compounds are mentioned in the manuscript, but only as the ingredients of the flowers, whereas their presence in fruits is well known and of great interest according to their anticancer and anti-inflammatory activities (Gleńsk et al., E. Triterpenoid acids as important antiproliferative constituents of European elderberry fruits. Cancer 2017, 69, 643–651; and the recent study on the occurrence of triterpenoids and sterols in elderberry-derived foods: Onolbaatar et al. Fruit and fruit-derived products of selected Sambucus plants as a source of phytosterols and triterpenoids. Plants 2025, 14, 1490 - these references could be cited in the manuscript). The presence of triterpenoids, including triterpenoid acids, requires attention because these compounds can participate in cardioprotective activity by reducing oxidative stress, inhibiting apoptosis, and improving heart function. The Author mentioned that polyhydroxylated triterpenoid acids were regarded as probably the major effective compounds of extracts tested in one of the cited studies.

Response: I have added more information about it: “Recently, Onolbaatar et al. [2025] have compared the phytosterol and triterpenoid profiles of fruits and fruit-derived products of selected Sambucus plants, including S. nigra and S. racemose. They observed that elderberry fruits are rich sources of phytosterols, espe-cially sitosterol (which represented up to 88% in red elder S. racemose), and triterpenoids (13.81 mg/g in S. nigra f. porphyrophylla cultivar Black lace “Eva”). The phytochemical analysis showed that ursolic acid and oleanolic acids are the main triterpenoids in the ex-tract from S. nigra fruits [Glensk et al., 2017; Onolbaatar et al., 2025]. Among various fruit-derived products (jam, juice, syrup, and wine), jams had the highest concentration of phytosterols and triterpenoids (a total of 340 µg/g) [Onolbaatar et al., 2025].

Reviewer 2 Report

Comments and Suggestions for Authors

The article, "Comparative Cardioprotective Potential of Various Parts of Sambucus nigra L., Sambucus williamsii Hance, and Their Products," is an interesting and valuable source of insights into the cardioprotective potential of these species. Before acceptance, I would like to offer the following suggestions:

  1. A paragraph discussing the pathophysiology of cardioprotection, including key target genes and molecular mechanisms, should be added to the introduction. This would establish the biological foundation for understanding how nigra and S. williamsii exert their cardioprotective effects, improving the clarity and scientific depth of the manuscript.
  2. The authors mention that no time criteria were applied to the search, but it is unclear why only 109 out of the 276 identified articles were included. Clarifying the specific selection criteria for these 109 articles—such as relevance, study quality, or experimental models—would strengthen the transparency of the review.
  3. The discussion on toxicity effects and biosafety would be more logically placed after the cardioprotective effects section. First, establishing the benefits of the berries would allow the reader to better understand the pros and cons before considering safety concerns. This would improve the flow and coherence of the review.
  4. Heading 5 ____ Reviewing the literature indicates that various plant-based food products, such as tea, cocoa, and coffee, exhibit cardiovascular protective effects through multiple mechanisms [78-81]. However, the authors should place greater emphasis on berry-specific studies when linking Sambucus (European and Chinese elderberry) species to cardioprotection, particularly in relation to antioxidant activity and modulation of the lipid profile. For example, studies such as, “Lyciumbarbarum's diabetes secrets: A comprehensive review of cellular, molecular, and epigenetic targets with immune modulation and microbiome influence", which explore cellular, molecular, and epigenetic targets, highlight the metabolic effects of goji berries, including antioxidant and lipid-modulating actions, which are crucial for understanding their potential in both diabetes management and cardioprotection. As the authors did not address genetic or cellular mechanisms, it would be more appropriate to prioritize berry-specific mechanistic studies early in the review, rather than focusing on plant products like coffee or tea, whose chemical structures and toxicity profiles are distinct from those of berries.
  5. In conclusion, the claim that nigra fruits, flowers, and fruit juice may be good candidates for functional foods with cardioprotective properties should be framed within the context of future well-established long-term clinical trials. Emphasizing the need for clinical validation and addressing toxicity concerns would provide a more comprehensive perspective on their potential use.
  6. The authors should include a section on future research directions in the review. Highlighting gaps in current knowledge and proposing potential areas for future studies would enhance the review's contribution to the field and guide upcoming research in this area.

Author Response

The article, "Comparative Cardioprotective Potential of Various Parts of Sambucus nigra L., Sambucus williamsii Hance, and Their Products," is an interesting and valuable source of insights into the cardioprotective potential of these species. Before acceptance, I would like to offer the following suggestions:

Response: Thank you for reviewing the manuscript and providing such helpful comments. All of them have been taken into consideration when revising the manuscript.

  1. A paragraph discussing the pathophysiology of cardioprotection, including key target genes and molecular mechanisms, should be added to the introduction. This would establish the biological foundation for understanding how nigra and S. williamsii exert their cardioprotective effects, improving the clarity and scientific depth of the manuscript.

Response: I have added short information about it (in the chapter of Introduction): “ Therefore, the first time, the aim of the present review is to provide and compare an overview of the cardioprotective potential of various parts not only of S. nigra L., but also of S. williamsii Hance, and their products in various models. Moreover, cardioprotective mechanisms of their main chemical constituents were demonstrated in this paper to provide a basis for further study and development. Potential molecular mechanisms of cardioprotection by chemical compounds from various parts of S. nigra L. may include its antioxidant activity (measured by different markers of oxidative stress), anti-inflammatory activity (by decreasing level of nitric oxide, and interelukin-2), lowering blood pressure (acting as renin inhibitor), hypolipidemic activity, and anti-obesity potential (by increasing hepatic AMP-activated protein kinase).” In addition, there is more information about in the chapter – “Cardioprotective potential of various parts of S. nigra, S. williamsii, and their products (in vitro and in vivo models)” and on Figure 1.

  1. The authors mention that no time criteria were applied to the search, but it is unclear why only 109 out of the 276 identified articles were included. Clarifying the specific selection criteria for these 109 articles—such as relevance, study quality, or experimental models—would strengthen the transparency of the review.

Response: I have added more information about it: “The data extracted from each article were: the type of studied material (species, cultivar, solvents used for extraction, or other relevant data), methods used in the study/study design, number of replicates, the cardioprotective activity, and statistical significance. If the P values were not provided for individual data points, it was assumed that the results were significant, unless it was stated otherwise in the text it.”

  1. The discussion on toxicity effects and biosafety would be more logically placed after the cardioprotective effects section. First, establishing the benefits of the berries would allow the reader to better understand the pros and cons before considering safety concerns. This would improve the flow and coherence of the review.

Response: I have corrected it. Now, it is: “4. Cardioprotective potential of various parts of S. nigra, S. williamsii, and their products (in vitro and in vivo models); “5. Toxic action and the bioavailability of bioactive compounds”.

  1. Heading 5 ____ Reviewing the literature indicates that various plant-based food products, such as tea, cocoa, and coffee, exhibit cardiovascular protective effects through multiple mechanisms [78-81]. However, the authors should place greater emphasis on berry-specific studies when linking Sambucus (European and Chinese elderberry) species to cardioprotection, particularly in relation to antioxidant activity and modulation of the lipid profile. For example, studies such as, “Lyciumbarbarum's diabetes secrets: A comprehensive review of cellular, molecular, and epigenetic targets with immune modulation and microbiome influence", which explore cellular, molecular, and epigenetic targets, highlight the metabolic effects of goji berries, including antioxidant and lipid-modulating actions, which are crucial for understanding their potential in both diabetes management and cardioprotection. As the authors did not address genetic or cellular mechanisms, it would be more appropriate to prioritize berry-specific mechanistic studies early in the review, rather than focusing on plant products like coffee or tea, whose chemical structures and toxicity profiles are distinct from those of berries.

Response: I have added more information about it: “Reviewing the literature indicate that various plant food products, like tea, cocoa, coffee, and other exhibit cardiovascular protective action using many mechanisms of action [78-81]. Different health benefits associated with the consumption of berries also demonstrated in epidemiological effects on CVDs. For example, various berries (such as sea buckthorn berries (Hippophae rhamnoides), aronia berries (Aronia melanocarpa), goji berry (Lycium barbarum), and other; in forms: fresh berries, wine, juice, teas and supplements) may play an important role in the modulating hemostasis, including blood platelet functions [Olas, 2017; Shunkai et al., 2025]. However, the cardioprotective potential of berries are dependent on a range of factors, such as the type of berries, the form of consumption, and their bioactive composition, especially phenolic compounds: anthocyanidins, procyanidins, flavonols, and phenolic acids [Cheng et al., 2010]. For example, these phenolic compounds may modulate blood platelet function by modifying the expression of blood platelet receptors, and changing the activity of signaling enzymes. They also change the level of ROS [Olas, 2018]. More details about implication of berries for human health are described in few review papers [Olas 2017 and 2018; Shunkai et al., 2025; Thorakkattu et al., 2025].”

  1. In conclusion, the claim that nigra fruits, flowers, and fruit juice may be good candidates for functional foods with cardioprotective properties should be framed within the context of future well-established long-term clinical trials. Emphasizing the need for clinical validation and addressing toxicity concerns would provide a more comprehensive perspective on their potential use.

Response: I have added more information about it: “S. nigra fruits, flowers, and fruit juice may be good candidates for functional foods with cardioprotective properties. However, well-established long-term clinical in this context trials should be a hot topic in the future. In addition, the clinical validation and addressing toxicity concerns would provide a more comprehensive perspective on their potential use.”

  1. The authors should include a section on future research directions in the review. Highlighting gaps in current knowledge and proposing potential areas for future studies would enhance the review's contribution to the field and guide upcoming research in this area.

Response: I have not decide to add new chapter, because I have already described this information in the chapter of Conclusion.

Round 2

Reviewer 1 Report

Comments and Suggestions for Authors

The manuscript was improved according to my suggestions. I have no more comments.